# ‘Garlic-lipo’4Plants: Liposome-Encapsulated Garlic Extract Stimulates ABA Pathway and PR Genes in Wheat (*Triticum aestivum*)

**DOI:** 10.3390/plants12040743

**Published:** 2023-02-07

**Authors:** Barbara Kutasy, Márta Kiniczky, Kincső Decsi, Nikoletta Kálmán, Géza Hegedűs, Zoltán Péter Alföldi, Eszter Virág

**Affiliations:** 1Department of Plant Physiology and Plant Ecology, Georgikon Campus, Institute of Agronomy, Hungarian University of Agriculture and Life Sciences, Festetics Str. 7, 8360 Keszthely, Hungary; 2Research Institute for Medicinal Plants and Herbs Ltd., Lupaszigeti Str. 4, 2011 Budakalász, Hungary; 3Department of Biochemistry and Medical Chemistry, University of Pécs Medical School, Szigeti Str. 12, 7633 Pécs, Hungary; 4Department of Information Technology and Its Applications, Faculty of Information Technology, University of Pannonia, Gasparich Str. 18, 8900 Zalaegerszeg, Hungary; 5EduCoMat Ltd., Iskola Str. 12/A, 8360 Keszthely, Hungary; 6Institute of Metagenomics, University of Debrecen, Egyetem Square 1, 4032 Debrecen, Hungary; 7Department of Environmental Biology, Georgikon Campus, Hungarian University of Agriculture and Life Sciences, Festetics Str. 7, 8360 Keszthely, Hungary; 8Department of Molecular Biotechnology and Microbiology, Institute of Biotechnology, Faculty of Science and Technology, University of Debrecen, Egyetem Square 1, 4132 Debrecen, Hungary

**Keywords:** Illumina sequencing, *Allium sativum*, genome-wide transcriptional profiling, biostimulant, garlic, abscisic acid, pathogenesis-related genes

## Abstract

Recently, environmentally friendly crop improvements using next-generation plant biostimulants (PBs) come to the forefront in agriculture, regardless of whether they are used by scientists, farmers, or industries. Various organic and inorganic solutions have been investigated by researchers and producers, focusing on tolerance to abiotic and biotic stresses, crop quality, or nutritional deficiency. Garlic has been considered a universal remedy ever since antiquity. A supercritical carbon dioxide garlic extract encapsulated in nanoscale liposomes composed of plant-derived lipids was examined as a possible PB agent. The present study focused on the characterization of the genes associated with the pathways involved in defense response triggered by the liposome nanoparticles that were loaded with supercritical garlic extracts. This material was applied to *Triticum aestivum* in greenhouse experiments using foliar spraying. The effects were examined in a large-scale genome-wide transcriptional profiling experiment by collecting the samples four times (0 min, used as a control, and 15 min, 24 h, and 48 h after spraying). Based on a time-course expression analysis, the dynamics of the cellular response were determined by examining differentially expressed genes and applying a cluster analysis. The results suggested an enhanced expression of abscisic acid (ABA) pathway and pathogenesis-related (PR) genes, of which positive regulation was found for the AP2-, C2H2-, HD-ZIP-, and MYB-related transcription factor families.

## 1. Introduction

Garlic (*Allium sativum* L., *Amaryllidaceae*) is a well-known medicinal herb with a remarkable therapeutic repute of providing many beneficial bioactive compounds [1,2,3]. Among the phyto-components of garlic, there is a high level of fructans [4], flavonoids [5], phenols, and polyphenol content [6,7,8]. The antimicrobial effect of garlic was reported in in vitro experiments that showed an effective response against many plant-pathogenic fungi and bacteria [9,10,11,12,13]. Plant-based extracts have several advantages for crops, such as increased resistance to abiotic and biotic stress and the possibility of reduced application of pesticides and mineral fertilizers [14,15]. Plant biostimulants (PBs) could be effective for achieving these aims by using diverse natural substances that generate physiological and molecular processes in plants to enhance nutritional efficiency, abiotic stress tolerance, and crop quality attributes [16,17]. There are several studies using biostimulants with different chemical and biological ingredients, but there is llimited literature about the use of extracts from garlic as PBs. Hayat et al. used aqueous garlic extracts (AGEs) as a biostimulant for foliar spray and fertilizationto improve crop quality and to promote the physiological potential of cucumber, tomato, eggplant, and pepper seedlings. Moreover, the AGEs exerted priming effects and defense responses against pathogenic fungal infections. This defense mechanism probably activated reactive oxygen species and antioxidant enzymes, such as superoxide dismutase (SOD), and peroxidase (POD) [18,19,20,21]. The application of AGEs has improved the growth and physiological efficiency of fava bean [22] and snap bean plants, and the level of endogenous phytohormones, particularly auxins, gibberellins, and salicylates [23].

Hormonal balance is another factor that has large effects on dormancy release, organ formation, and development. The role of hormones in the dormancy release and sprouting of garlic cloves has already been demonstrated [24,25]. These studies suggested that abscisic acid (ABA) played a major role in the whole differentiation/growth process of garlic plants. Garlic contains large amounts of the compounds described above; therefore, we hypothesize that garlic may have on significant effect when applied to wheat plants and stimulate the physiological processes associated with these compounds. This assumption was examined in this study, which aimed to analyze these relevant processes.

Extraction techniques using supercritical fluids may create greener processes and end products than conventional alternatives [26], leading to efficient solvents with better transport properties (diffusivity, mass transfer coefficient, and penetration ability) than many liquid organic solvents [27]. Supercritical extracts of algae as biostimulants were comprehensively investigated in winter wheat, rapeseed, and mustard [28,29]. Hincapié et al. (2008) [30] reported that using supercritical carbon dioxide (SC-CO_2_) as a solvent produced garlic extracts, which were tested on *Tetranychus urticae*. The SC-CO_2_ garlic extracts do not contain any toxic extraction solvents and preserve oxidation-prone substances because they are not exposed to oxygen or high temperatures during the extraction process [31]. To date, the use of SC-CO_2_ garlic extracts as PBs in wheat has not been reported.

The nanoscale drug delivery system is a technology used extensively in medicine to increase drug efficiency and to reach target tissues. These modern nanotechnological solutions are considered in agronomy as well [32,33,34]. Karny et al. (2018) [35] reported the use of liposomes composed of plant-derived lipids that penetrated the leaves and translocated to other leaves and to the roots of tomato plants, where active compounds were released. The efficiency of penetration of the applied nanoparticles was up to 33 percent, compared to 0.1 percent of crop-protection agents in a similar treatment. Moreover, airborne nanoparticles disintegrated into safe molecular building blocks (phospholipids) over longer airborne distances to reduce the environmental toll. An example is the liposome-encapsulated plant conditioner product, Elice16indures^®^ of RIMPH Ltd., Budakalász, Hungary (containing 11 herbal CO_2_ extracts involving garlic), and in both phytotron and earlier field experiments, it was found that this agent triggered the plant defense system through hormonal pathways in *Hordeum vulgare* [36] and *Glycine max* [37,38].

In this study, the effect of SC-CO_2_ garlic extract encapsulated in nanoscale liposomes (preliminarily named as ‘Garlic-lipo’) as a PB was examined. The mechanism of action was investigated by using genome-wide transcriptional profiling of genes associated with the defense response triggered by the ‘Garlic-lipo’. The results presented here are based on the QuantSeq 3′ mRNA sequencing experiments, which data have been previously reported by Kutasy et al. (2022) [39].

## 2. Materials and Methods

### 2.1. Garlic-Lipo Stability and Particle Size Measurements of Liposomes

We tested the ‘Garlic lipo’ agent as an active component of ELICE Vakcina^®^, a member of the Elice16 product line developed in the Research Institute for Medicinal Plants and Herbs (RIMPH) Ltd., Hungary (https://gynki.hu/en/rimph-botanicals/products/ accessed on 1 January 2023), and which formulation and biological effect have already been characterized [36,38]. According to the formulation technology of ELICE Vakcina, the supercritical fluid extraction of fine-grated fresh garlic with natural carbon dioxide (FLAVEX Naturextrakte GmbH, Rehlingen-Siersburg, Germany) was encapsulated in sunflower lecithin-based liposomes.

The electrostatic stabilization, dispersion properties, and nanoparticle size distribution were measured by dynamic light scattering using the Zetasizer Ultra instrument at room temperature (Malvern Panalytica, Malvern, UK). The polydispersity index (PI) and Zeta potential were measured in four technical repeats. The dilution of the samples was 10^−1^ each time using Milli-Q water. The diluted suspension was placed in a disposable, low-volume cuvette with a path length of 10 mm (Malvern Instruments, Malvern, UK). The size distribution of each liposome sample and the particle size mode were measured using detectors at 3 different angles to account for the front (13°), side (90°), and back (173°) scatter of light at room temperature. Each sample was measured at each angle in triplicate, with adaptive correlation applied to each, to improve overall data quality. The data were processed using the ZS Xplorer Software V.2.3.1 The particle size of the liposomes is given from the cumulative analysis of four measurements of the average Z-diameter and PI.

### 2.2. Plant Materials and Treatment with ‘Garlic-Lipo’

The seeds of *T. aestivum* cultivar ‘Cellule’ (France, 2012) were germinated in Petri dishes at 30 °C/20 °C with a 16 h photoperiod, and they were planted in pots and grown under controlled greenhouse conditions. The plants were sprayed with ‘Garlic-lipo’ in a dosage of 240 g·ha^−1^ using a Euro Pulvé plot sprayer at the BBCH12 stage [40]. Fresh leaves were collected before the treatment as control (0) and at 15 min, 24 h, and 48 h after treatment (Figure 1). The samples were stored in a RNA Shield (Zymo Research, Irvine, CA, USA) at −25 °C until RNA analysis. Three biological repeats were used for further analyses.

### 2.3. Sequencing, Data Processing, Gene-Level Quantification, and DEG Determination

The Illumina Gene Expression Profiling (GEx) library preparation, RNA sequencing, sequence read pre-processing, gene-level quantification, and differentially expressed gene (DEG) analysis were performed as described in our previous papers [39,41]. The RNA libraries were sequenced with a final output single-end, with 14–26 M x 85 bp long reads. The raw reads were deposited in the National Center for Biotechnology Information (NCBI) database (as the BioProject PRJNA808851) and the Sequence Read Archive (SRA) database (SRR18107544, SRR18107543, SRR18107542, and SRR18107541). A de novo reference transcript was created using these SRA datasets to reconstruct transcripts that represent the experimental samples. This transcript dataset was deposited in the Transcriptome Shotgun Assembly (TSA) database at DBJ/EMBL/GenBank under the accession GJUY00000000.1. To estimate transcript abundances, CountTable was created [39], which is accessible in Mendeley Data (https://data.mendeley.com/datasets/p66v4yxbtp/1, accessed on 17 March 2022). DEGs were calculated based on the CountTable by applying a pairwise differential expression analysis as described by Hegedűs et al. (2022) [41].

### 2.4. Time-Course Expression Analysis

A time-course expression analysis of the count data arising from the RNA-seq was performed to detect genomic features with significant temporal expression changes and with significant differences between the experimental groups [39]. The genes with significantly different expression levels were divided into nine clusters. The TimeCourse Table is accessible in Mendeley Data (https://data.mendeley.com/datasets/xvvscxpz6w/1, accessed on 18 March 2022). According to the expression dynamics of the changes of the clustered genes, these nine clusters were classified into four groups: (1) expression level gradually increased; (2) expression level gradually decreased; (3) expression level initially increased and then reduced; and (4) expression level initially decreased and then returned to its original level. The clusters were annotated with Gene Ontology (GO) terms and visualized [39]. Group 1 genes with gradually increased expression levels were further examined and discussed in this study.

### 2.5. Functional Annotation

The methods of functional annotation and GO analysis were described in a previous publication [37]. The annotation of the entire transcriptome is presented in the AnnotationTable that is accessible in Mendeley Data (https://data.mendeley.com/datasets/p66v4yxbtp/1, accessed on 17 March 2022) and described in Kutasy et al. (2022) [39].

### 2.6. Real-Time PCR Analysis

Total RNA was quantified using a Nanodrop 2000c spectrophotometer and a Qubit 2.0 fluorometer (Thermo Fischer Scientific, Waltham, MA, USA). Five hundred nanograms (ng) of total RNA was reverse transcribed using the M-MuLV RT (Maxima First Strand cDNA Synthesis Kit, Thermo Fischer Scientific, Waltham, MA, USA). A total of 50 ng of cDNA was used in 10 μL reactions for real-time PCR using the Xceed qPCR SG 2× Mix (Institute of Applied Biotechnologies, Praha-Strašnice, Czech Republic) and a CFX384 Touch Real-Time PCR Detection System (Bio-Rad, Hercules, CA, USA). The data (Appendix A) were analyzed using the ΔΔCt method [42]. Tubulin and ubiquitin expressions were used as a reference for gene expression.

## 3. Results

To date, the effect of garlic extract was investigated in vegetable plants in studies that focused on yield quality and physiological function, as well as stress responses. However, it has not yet been examined in monocotyledonous plants. Moreover, the effect of garlic extract on changes in gene expression levels has not been studied.

High levels of flavonoids, phenolics, fructans, and abscisic acid hormone (ABA) were found in garlic. Therefore, in this study, the processes related to these compounds were examined, which might be stimulated by garlic extract.

This study presents the effects of garlic SC-CO_2_ extract on wheat plants by investigating ‘Garlic-lipo’. The formulation of this agent was carried out with a focus on better uptake by plants after treatment, with a strict analytical control measuring the size of the nanoparticle and the Zeta potential. The mechanism of this effect was investigated using transcriptome analysis, such as mRNA-sequencing, determination of DEGs, and time-course expression analysis, showing high expression changes in response to the treatment. The in silico analysis of mRNA sequencing and gene expression was validated using RT-qPCR methods.

### 3.1. Dynamic Light Scattering

The size determination parameters are summarized in Table 1, and representative figures of the measurements are shown in Figure 2. The Zeta potential analysis is summarized in Table 2 and Figure 3. The liposome properties of the investigated samples indicated that either the particle size or the polydispersity index (*p* > 0.05, Pearson) was relatively constant (149.2 nm in size, polydispersity index = 0.13035, and *n* = 4) in the four measured samples.

### 3.2. Pairwise Expression Analysis and Determination of Top 50 DEGs

The expression levels of DEGs measured at three sampling times (15 min/24 h/48 h) after the ‘Garlic-lipo’ treatment were compared to the control (0 min) for the largest difference in the top 50 genes presented in the heatmaps (Figure 4, Appendix A). Many differentially expressed (DE) features were detected in the 15 min sample [(probability > 0.9): 1712; up-regulated (M > 0): 919; down-regulated (M < 0): 793]; in the 24 h sample [(probability > 0.9): 2485; up-regulated (M > 0): 1279; down-regulated (M < 0): 1206]; and in the 48 h sample [(probability > 0.9): 2578; up-regulated (M > 0): 865; down-regulated (M < 0): 1713].

Proteins involved in stress response and the ABA pathway showed up-regulated levels in the 15 min sample (Appendix A), such as ABA-responsive protein HVA22, late embryogenesis abundant protein (LEA), 9-cis-epoxycarotenoid dioxygenase (NCED3), and U-box domain-containing protein 19 (PUB19). The down-regulated genes were found to be associated with photosynthesis, as well as ribosomal RNA genes.

The top 50 DEGs were mostly down-regulated in the 24 h sample (Appendix A), except for pathogenesis-related protein 1 (PR1) and aldehyde dehydrogenase family 7 member A1 (ALDH7). Photosynthesis-related and ribosome-related genes were shown to be down-regulated. The largest difference was found in the 48 h sample, which is presented in Figure 4.

While there were several ABA-related genes determined in the 15 min sample and the 24 h sample, only one was found in the 48 h sample. As for up-regulated genes, the expression level of flavanone 3-hydroxylase (F3H) gene associated with the ABA pathway was increased. Moreover, PR1-5 genes and other stress-induced genes were proven to be up-regulated. The up-regulated PR genes, such as PR1, endo-beta-1,3-glucanase (PR2), chitinas (PR3), wheatwin-2 protein (PR4), and thaumatin protein (PR5), were significantly different in their expressions. One of the PRs, the Bowman–Birk-type trypsin inhibitor (PR6), showed a down-regulated expression level.

Biotic and abiotic stress response genes, such as alpha-amylase/trypsin inhibitor (ATI), caseinolytic protease B1 (ClpB1), wheat-induced resistance 1B (WIR1B), and heavy metal-associated isoprenylated plant protein 27 (HIPP27), were up-regulated in the sample of 48 h. Different transferases, such as UDP-glucose flavonoid 3-O-glucosyltransferase 7 (GT7), tricetin 3′,4′,5′-O-trimethyltransferase (OMT2), and acetylserotonin O-methyltransferase 1 (ASMT), were shown to be up-regulated level in the same sample.

Moreover, biotic and abiotic stress-responding transcription factors (TFs) that consist of HD-ZIP, C2H2, and AP2 families, such as zinc finger protein 12 (ZAT12—C2H2 family), homeobox-leucine zipper transcription factors 22 and 24 (HOX22 and HOX24—HD-ZIP family), dehydration-responsive element-binding protein 1G (DREB1G—AP2 family), and MYBS1—MYB-related family, were observed in each sample evaluated at all of the three timepoints of this study.

The determination of the top 50 DEGs demonstrated the changes in the expression levels of ABA pathway-related genes, PRs, and TF genes (Figure 5). To obtain more information about the physiological processes in wheat after the biostimulation treatment applied here, the samples were examined using a time-course expression analysis.

### 3.3. Time-Course Expression Analysis

A time-course expression analysis was an appropriate tool for evaluating the data arising from the RNA-seq technology to represent genes with significant changes in their temporal expressions and with significant differences between the experimental groups. Using TSA and CountTable data, 5287 contigs were tested and filtered, which resulted in nine clusters. The clusters were classified into four groups according to their dynamics of changes: those of which the expression level (i) gradually increased, (ii) gradually decreased, (iii) initially increased and then reduced, or (iv) initially decreased and then returned to its original level. The annotated genes were examined by using the GO terms and biological processes that were already presented in a previous paper (Kutasy 2022). The group of genes with expression levels that gradually increased ((i) cluster 1) was shown mainly to be associated with defense response processes, while the groups of genes with expression levels that gradually decreased ((ii) clusters 3, 4, 6, and 7) were shown to have down-regulated cellular biosynthetic and metabolic processes. Moreover, the groups of genes with expression levels that initially increased and then reduced ((iii) clusters 2, 8, and 9) were mostly shown to be involved in different transport and translation processes. The group of genes with expression levels that initially decreased and then returned to their original levels ((iv) cluster5) was associated with RNA-related (transcription) and metabolic processes (Kutasy 2022). The genes of the nine clusters were also classified using the Kyoto Encyclopedia for Genes and Genomes (KEGG) analysis (Appendix A). In this study, the genes with expression levels that gradually increased showed defense response associations (Figure 6).

The time-course analysis identified 24 pathways, with a total of 201 sequences (Appendix A) in (i) cluster 1, and enzymes that may be involved in primary defense reactions. Regarding defense response, plant–pathogen interaction; mitogen-activated protein kinase (MAPK) signaling pathway; plant hormone signal transduction; flavonoid, terpenoid, and phenylpropanoid biosynthesis; glutathione metabolism; and metabolism by cytochrome P450 were found to be the most important pathways (Figure 6). The disease-resistance protein PR1 was involved in the pathways of plant–pathogen interaction (map04626), plant hormone signal transduction (map04075), and MAPK signaling (map04016), while the RPM1 protein was found to be associated with the plant–pathogen interaction (map04626) pathway. The mono- and diterpenoid biosynthesis (map00902, 00904) pathways were identified in all of the samples, involving the proteins of 7-deoxyloganetin glucosyltransferase (UGT85A23-24), ent-kaurene oxidase (CYP701A), and ent-sandaracopimaradiene 3-hydroxylase (CYP701A8). In the phenylpropanoid biosynthesis (map00940) pathway, the proteins of shikimate O-hydroxycinnamoyl-transferase (HST), caffeic acid 3-O-methyltransferase/acetylserotonin O-methyltransferase (COMT), and peroxidase were found. The cytochrome P450 and glutathione metabolic (map00980, 00480) pathways were also identified, involving the proteins of glutathione S-transferases (GST23 and GSTU6) and L-ascorbate peroxidase (APX).

Based on the time-course expression analyses of (ii) clusters 3, 4, 6, and 7, the down-regulated cellular biosynthetic and metabolic processes may be involved in photosynthesis-associated processes; the processes of peroxisome, glyoxylate, and dicarboxylate metabolisms; linoleic acid metabolism; oxidative phosphorylation; and ribosomal processes. Regarding the plant–pathogen interaction pathway, heat shock proteins were observed.

The time-course analysis of (iii) clusters 2, 8, and 9 showed enzymes that may be involved in process of endocytosis; MAPK signaling pathway; and cysteine and methionine metabolisms. These clusters are also associated with ribosomal processes in the cytosol.

Moreover, the time-course analysis of (iv) cluster 5 was associated with mRNA processing, RNA phosphodiester-bound hydrolysis, and nucleic acid metabolic processes. The KEGG analysis of cluster 5 gave no results.

### 3.4. Real-Time PCR Analysis

The most important defense response-relevant genes of the top 50 DEGs were examined using real-time PCR. PR1-5 showed an increased expression level during the whole experimental period as it was the highest in the 48 h sample—as it was observed in the transcriptome analysis. The genes associated with the ABA pathway (NCED3, HVA22, PUB19, and LEA31) and TFs (HOX24, ZAT12, and MYBS1) also showed similar changes in their expression levels as revealed in the transcriptome analysis. Typically, these genes showed maximal expression levels in the 15 min sample and the 24 h sample. Moreover, a TF (DREB1G) was prognosticated to have the highest expression level in the 48 h sample in the DEG analysis, and this was confirmed by the real-time PCR (Figure 7 and Appendix A).

## 4. Discussion

Worldwide, the scientific community and commercial enterprises have focused on the development of environmentally safe methods that facilitate plant growth and crop protection. PBs could be effective chievinging these aims as they are diverse natural substances that generate physiological and molecular processes in plants to enhance nutritional efficiency, abiotic and biotic stress tolerances, and crop quality attributes [16,17,43]. Using plant-based extracts has several advantages, such as enhanced yields and quality of crops, low toxicity to humans and the environment, increased resistance of crops to abiotic and biotic stresses, and reduction in the application of pesticides and mineral fertilizers [14,15]. Aqueous garlic extract as a plant biostimulant was recently reported as a promising agent to trigger plant immunity [18]. Focusing on low-input sustainable agriculture, we aimed to reinforce the efficacy of garlic extract using SC-CO_2_ extraction and liposome nanotechnology for better uptake in and delivery to plants. A novel ‘Garlic-lipo’ production and its treatment on wheat plants were designed. The time-course effect of ‘Garlic-lipo’ was tested by using whole-genome transcriptional profiling and enzyme activity measurements. The up-regulated genes were validated by using RT-qPCR experiments.

The liposome size determination and the monitoring of colloid characteristics during the ‘Garlic-lipo’ production were controlled using dynamic light scattering. The liposome sizes were in the range between 134.2 and 164.2 nm, and the colloid stability proved to be high, showing a Zeta potential between −40.66 and −40.53 mV. These physicochemical properties of the nanoliposomes provide the product a high stability and great bioactivity [44]. However, there are only a few reports on the application of such a product in agricultural practice.

This product was tested using time-course experiments. The QuantSeq 3′ mRNA sequencing for RNA quantification in the samples had been performed earlier (data reported in Kutasy 2022 [39]). The determination of DEGs showed an increase in the expression levels of genes associated with the ABA pathway, which was observed especially 15 min after treatment. A parallel can be drawn with the remarkable amounts of endogenous phytohormones described in garlic clove extracts, among which a high level of ABA was found by Arguello (1991) [24]. The cultivar, photoperiod, and temperature affected the hormone levels when ABA showed up to 10–25-fold higher levels than cytokinins, auxins, and jasmonic acid [45]. A treatment with a high ABA-content garlic extract, therefore, may stimulate plant responses to biotic or abiotic stresses and increase yield levels, as was shown by previous exogenous ABA treatments [46,47,48]. Based on this information, it was hypothesized that a garlic extract with a high ABA content might stimulate ABA pathway-related genes. This theory was proven here in these experiments. In these experiments, we observed a rapid up-regulation of these genes at 15 min after the ‘Garlic-lipo’ treatment, but their expressions gradually decreased when the samples were taken 24 h after the treatment. Among the top 50 DEGs (15 vs. 0 min), the homologous protein of stress-and ABA-induced HVA22, LEA, and NCED3 is the key rate-limiting enzyme in the ABA biosynthetic pathway regulating multi-abiotic stress tolerance, as was found in [49,50,51,52,53]. The regulators of salt-tolerant [54] and negative regulators of drought stress-response protein [55] and PUB19 [56] were also up-regulated, which were also known to be sensitive to ABA in *Arabidopsis thaliana* [57]. The down-regulated genes were associated with photosynthesis, as well as ribosomal RNA genes. A down-regulation of photosynthetic genes by ABA signaling was observed by Staneloni et al. (2008) [58] and Gao et al. (2016) [59], and this phenomenon might be associated with ABA-mediated chlorophyll degradation and leaf senescence.

The regulation of ABA-signaling genes was associated with the overexpression of TFs, with their highest values observed at 15 min. TFs are members of the HD-ZIP, bZIP, C2H2, and AP2 families, such as ZAT12 (C2H2 family), that participate in the induction and repression of cold-responsive and oxidative stress genes [60]; HOX22 (HD-ZIP family) is involved in the defense mechanism of pathogen-related stress response of fungi [61], and drought- [62] and heat-stress-tolerance-induced HOX24 [63] and MYBS1 [64] are the essential components of the sugar signaling pathway regulating the α-amylase gene promoter [65]. A high expression level was found for DREB1G (AP2 family), which is a typical C-repeat-binding factor (CBF/DREB1) whose specific functions was found to be related to cold stress response [66,67]. Interestingly, this TF was the only one present when the sample was taken 48 h after the treatment. A microarray analysis using Arabidopsis cultured cells revealed that several ABA-inducible genes and the TF ZAT12 are induced by oxidative stresses, which might be a regulator in the reactive oxygen species (ROS) scavenging mechanism that is involved in multiple stresses, such as wounding, pathogen infection, and abiotic stresses [68].

The CBF cold-responsive pathway includes the three key TFs of CBF1, CBF2, and CBF3, and this pathway also plays a significant role in the cold acclimation of *Arabidopsis* [69]. The CBF proteins as transcriptional activators recognize both the C-repeat (CRT) and the DREB. It was shown that the overexpression of CBF2 and CBF3 can induce several cold-responsive (COR) genes [70,71].

Vogel et al. (2005) [60] reported that ZAT12 was also induced with CBFs, which regulated the expression of 24 cold-responsive genes, indicating that ZAT12 might be involved in cold acclimation via a novel cold-response pathway [72,73].

The up-regulated PR genes were found both in the 24 h sample and in the 48 h sample after the ‘Garlic-lipo’ treatment, of which they showed a higher expression level in the 48 h sample. The top 50 DEGs of 48 vs. 0 h indicated significantly higher expressions of PR genes, such as PR1, which are known as the hallmarks of defense pathways in wheat [74,75]. PR2 is involved in the resistance response of wheat to stripe rust (*Puccinia striiformis* f. sp. *tritici*) [76,77]; PR3 constitutes the second largest group of antifungal proteins [78]; PR4 has antifungal activity against several pathogenic fungi (*Fusarium* sp., *Botrytis* sp.) [79,80]; and PR5 is mostly associated with responses to biotic stresses, especially drought and osmotic stresses [81]. Additionally, F3H, which is an ABA-induced abiotic stress gene that catalyzes the third step of the central flavonoid biosynthetic pathway [82,83], was found to be up-regulated in this sample. One of the PRs, the PR6, showed a down-regulated expression level [84,85].

To summarize, the majority of the top 50 DEGs might be grouped into (i) ABA-pathway and (ii) PR genes; however, their induction was strongly associated with time. A rapid response to the drug was observed in ABA signaling, but PR induction was detected after 24–48 h of the treatment. It was presumed that the regulation of these genes was under the control of (iii) TFs, which were not under strict time control, such as ABA and PR. The observed time-shift effect of PR genes is characteristic of priming inducers that might be triggered by fungus or chemical treatments [85,86]. The expression dynamics of PR1, PR3, PR4, PR5, PR6, NCED3, HVA22, PUB19, LEA 31, HOX24, DREB1G, MYBS1, and ZAT12 were validated by the RT-qPCR as well.

Additionally, biotic and abiotic stress-response genes were up-regulated in the 48 h sample, such as ATI, which has a defense mechanism against insect attack [87]; ClpB1, which is a high-molecular-weight chaperon protein that is part of the heat shock protein 100 (HSP 100) family [88]; WIR1B, which is a small glycine- and proline-rich protein having a key role in defense reactions against fusarium head blight (FHB) infection caused by the fungus *Fusarium graminearum* [89] and in the plant–pathogen interactions of *Blumeria graminis* [90]; and HIPP27, which is a metallochaperone that has been reported to play an important role in cadmium detoxification [91] and in response to nematode infection [92]. Different transferases showed up-regulated levels in the 48 h sample, such as GT7, which primarily transports sugar molecules to aglycones to enhance solubility and stability during flavonol biosynthesis [93]; OMT, which catalyzes three sequential methylations of flavone tricetin; and ASMT1, which is responsible for the syntheses of N-acetylserotonin and melatonin. Melatonin is synthesized from tryptophan and plays an important role in enhancing the resistance of plants against biotic and abiotic stresses [94]. Melatonin treatment significantly enhanced the drought tolerance of wheat seedlings by decreasing membrane damage [95]. The down-regulated genes were genes associated with photosynthesis and ribosomal RNA genes.

Since the effects of ‘Garlic-lipo’ showed a strong time factor relation, a time-course gene expression analysis was performed to determine other gene clusters induced by the treatment effects. This investigation resulted in nine clusters, in which the genes were annotated by determining their GO terms and using a KEGG analysis (Appendix A). Both the GO and KEGG analyses showed an increase in the expression levels of genes associated with defense response processes. During the experimental period, the genes involved in the biosynthesis of PR1-5 genes, terpenoids, and phenylpropanoids operated at a consistently high level. PRs have been found to be activated in response to different biotic and abiotic stresses, so PR1-5 proteins have a wide range of functions, such as protease inhibitor, glucanase, chitinase, and thaumatin protein [96]. Furthermore, the CYP450 and glutathione transferase genes involved in stress response showed a high level of expression. The GST genes are highly inducible by a wide range of stress conditions, including biotic stress [97].

A down-regulation of genes involved in photosynthesis was observed based on a heat map and KEGG analysis. For the entire period, the down-expression of the processes involved in photosynthesis was detected using time-course GO analysis, but it was not one of the most prominent processes (Figure 2 in Kutasy et al. (2022) [39]). Moreover, the KEGG analysis showed that the process of restoring photosynthesis had already begun. In the porphyrin and chlorophyll metabolism pathway (map00860) and the carbon fixation pathway in photosynthetic organisms (map00710), the expression of chlorophyllase (CLH) and malate dehydrogenase (MDH) were found to be up-regulated.

To summarize, the main part of the top 50 up-regulated DEGs affecting plant development and pathogen defense mechanism was found after the ‘Garlic-lipo’ treatment. These genes suggest a strong regulation of the ABA pathway and resistance mechanism. This phenomenon indicates physiological relations, with the synthesis of the plant hormone ABA having a key role in plant development, control of organ size, stomal closure, and signaling under different plant stresses. The ‘Garlic-lipo’ may rapidly affect the ABA signaling pathway in plants in response to nanoparticle exposure, leading to enhanced ABA-related gene activity and defense system. The assumed phytopathogen activity of this agent was also demonstrated to trigger PR genes involving systemic acquired resistance (PR1, PR2, and PR5) and local acquired resistance (PR3, PR4). Additionally, the common upregulated DEGs of the investigated samples found in the KEGG pathway analysis were involved in monoterpenoid, phenylpropanoid, diterpenoid, and stilbenoid biosynthesis, and glutathione and phenylalanine metabolism. These pathways have a key role in plant development, cell cycle, stem growth regulation, induced germination, defense mechanisms, and disease resistance (Figure 8). Based on all of these results, it can be concluded that the transcriptomic changes, which were examined using several approaches, unanimously indicate that the use of ‘Garlic-lipo’ may improve the health and development of plants by strengthening their response to biotic and abiotic stresses involving an emerging pathogen attack.

## 5. Conclusions

These results highlight the significant effects of the SC-CO_2_ garlic extract as the up-regulations of several defense-response genes were demonstrated. The increase in the expressions of the ABA pathway and PR genes was shown to be associated mainly with defense mechanisms. In conclusion, the plant’s innate immunity and defense response mechanisms might be triggered systemically by this agent, suggesting a potential priming active substance. Due to its extraction process and nanotechnological formulation, ‘Garlic-lipo’ may be a promising PB that can support sustainable agriculture. To substantiate the effects described here, we are planning to conduct further infection tests and field experiments.

## Figures and Tables

**Figure 1 plants-12-00743-f001:**
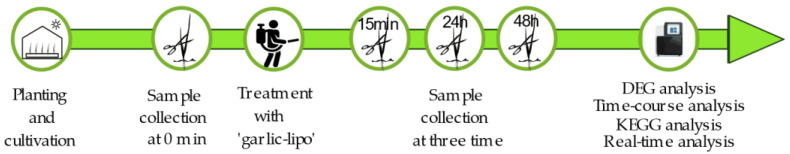
Sample collection at three timepoints of the nanoscale liposome-encapsulated SC-CO_2_ garlic extract (‘Garlic-lipo’) on *Triticum aestivum* in greenhouse experiments. The effects were examined in a large-scale genome-wide transcriptional profiling experiment collecting samples at various timepoints (0 min is the control, and 15 min, 24 h, and 48 h after spaying) and were evaluated by using DEG and time-course expression analysis.

**Figure 2 plants-12-00743-f002:**
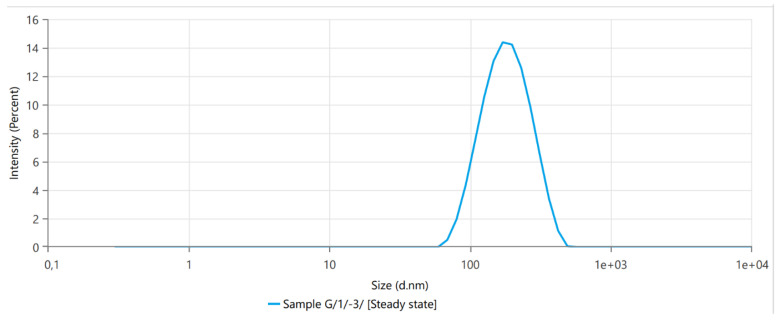
Size determination diagram of liposomes from the ’Garlic-lipo’ colloid. Liposome sizes are in the range between 134.2 and 164.2 nm.

**Figure 3 plants-12-00743-f003:**
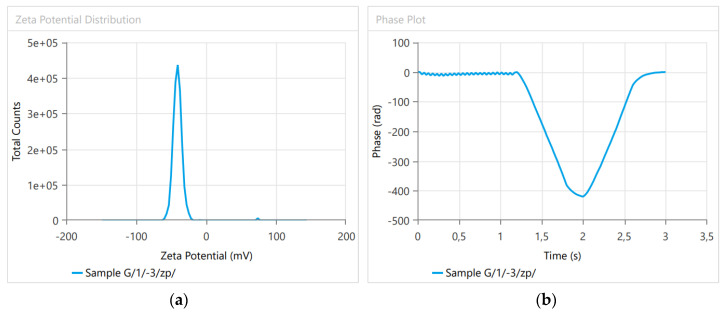
Determination of the Zeta potential based on the electrophoretic mobility of liposomes. (**a**) Zeta potential distribution: the colloid stability is proven to be high, showing the Zeta potential between −40.66 and −40.53 mV. (**b**) Phase plot to estimate the electrophoretic mobility of particles: Phase difference is proportional to the electrophoretic mobility of the nanomaterial and, thus to the Zeta potential of the aqueous medium.

**Figure 4 plants-12-00743-f004:**
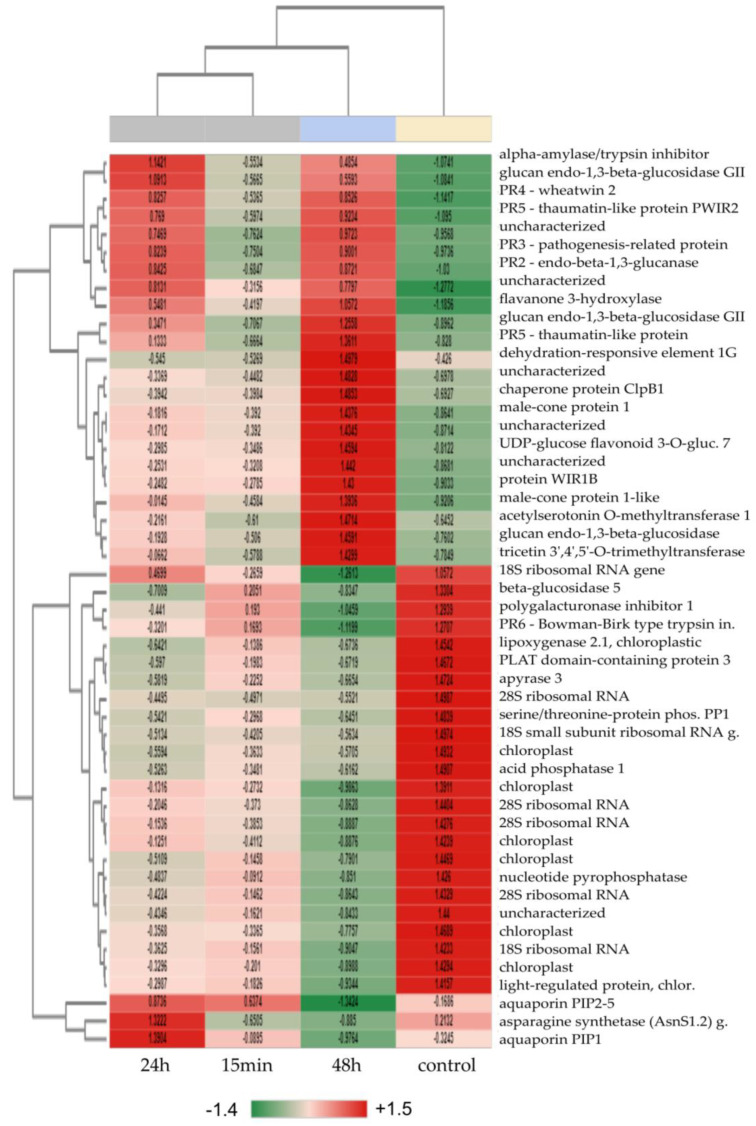
Heat map of the top 50 differentially expressed genes (DEGs) based on the comparison of the 48 h vs. control samples. The heatmap color range is from red for positive Z-score values to green for Z-score negative values. The number of differentially expressed (DE) features (Probability > 0.9): 2578. Up-regulated (M > 0): 865. Down-regulated (M < 0): 1713. For the annotation of the transcript IDs, see the Annotation Table (DOI: 10.17632/p66v4yxbtp.1).

**Figure 5 plants-12-00743-f005:**
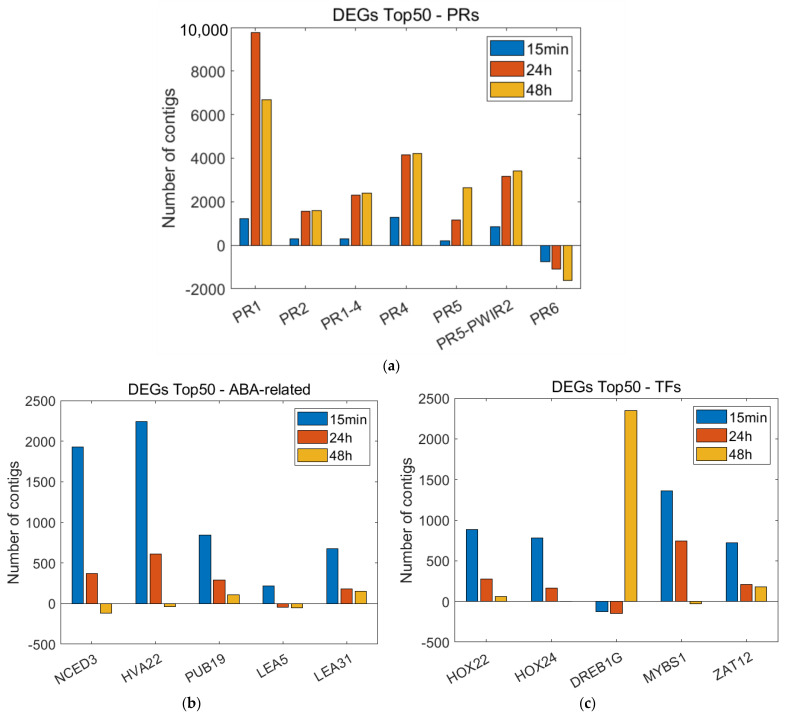
The determination of the top 50 differentially expressed genes (DEGs) in silico is represented in three sampling times. The numerical analysis of DEGs in a pairwise comparison of two times was examined using OmixBox.BioBam based on the RSEM and edgeR programs. These quantitative statistical methods to evaluate the significance of individual genes were implemented to examine the changes in the expression levels of (**a**) PR genes (control vs. 48 h); (**b**) ABA pathway-related genes (control vs. 15 min); and (**c**) TF genes (control vs. 15 min).

**Figure 6 plants-12-00743-f006:**
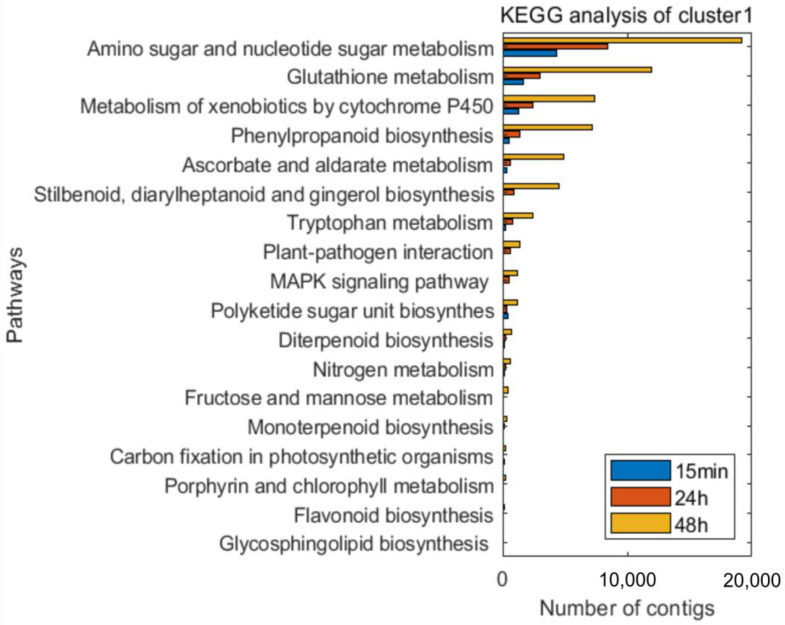
KEGG analysis of cluster 1 at the three investigated times, relative to the control (0 min). The expression levels of these genes gradually increase. The graph shows the numbers of transcripts that are annotated (DOI:10.17632/p66v4yxbtp.1) and can be placed into the KEGG pathway (Appendix A) using the TimeCourseTable (DOI: 10.17632/xvvscxpz6w.1).

**Figure 7 plants-12-00743-f007:**
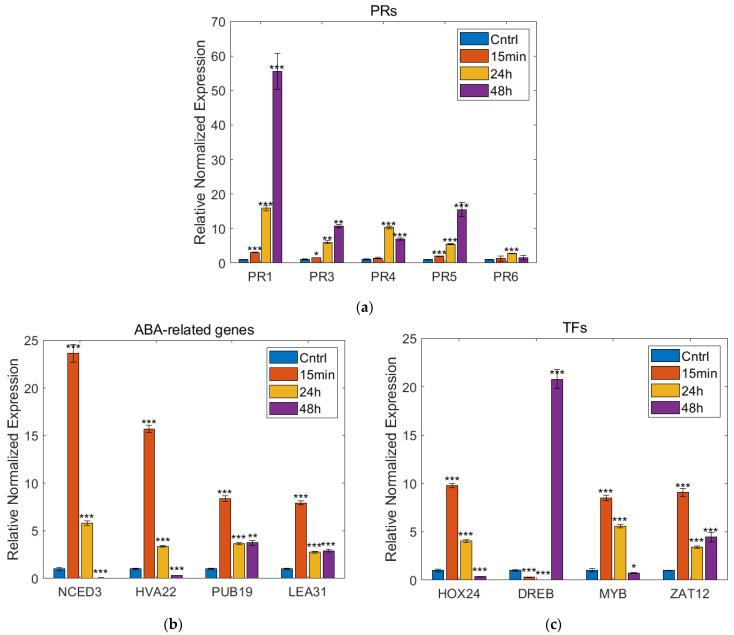
Results of the real-time PCR reactions performed at three sampling timepoints compared to control. The changes in the expression levels of (**a**) PR genes; (**b**) ABA pathway-related genes; and (**c**) TF genes. The values are represented as mean ± SEM (*n* = 4). *p* < 0.05 vs. Control, *p* < 0.05 vs. treated group. One asterisk indicates statistically significant difference between the means of a treated sample set compared to the mean of the control sample set to 5%; two asterisks indicate statistically significant difference to 1%; three asterisks indicate statistically significant difference to 0.1%.

**Figure 8 plants-12-00743-f008:**
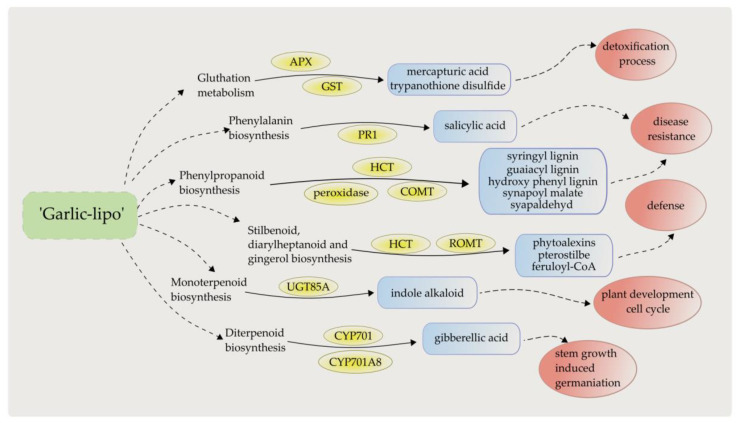
Summary of up-regulated DEGs (yellow) found in the KEGG pathways (not colored), with the patway-related secondary metabolites (blue), and their physiological roles (red) in defense mechanism and plant development as a response to ‘Garlic-lipo’ treatment.

**Table 1 plants-12-00743-t001:** Particle size distribution (PSD) by intensity statistics.

Name	Mean	Minimum	Maximum
Z-Average (nm)	149.2	134.2	164.2
Polydispersity Index (PI)	0.13035	0.1303	0.1304
Intercept	0.951	0.949	0.953
Peak One Mean by Intensity (nm)	160.6	129.6	191.6
Peak One Area by Intensity (%)	100	100	100
Derived Mean Count Rate (kcps)	171,300	1.71 × 10^5^	1.71 × 10^5^

**Table 2 plants-12-00743-t002:** Measurement of Zeta potential and quality of individual liposomes of the ’Garlic-lipo’ samples.

Name	Mean	Minimum	Maximum
Derived Mean Count Rate (kcps)	171,300	1.71 × 10^5^	1.71 × 10^5^
Zeta Potential (mV)	−40.595	−40.66	−40.53
Zeta Peak One Mean	−41.18	−41.43	−40.93
Zeta Peak Two Mean	−9.2	−9.25	−9.15
Conductivity (mS/cm)	0.007439	0.007399	0.007479
Wall Zeta Potential (mV)	−66.28	−66.33	−66.23
Zeta Deviation (mV)	8.632	8.607	8.657
Derived Mean Count Rate (kcps)	85,150	8.52 × 10^4^	8.52 × 10^4^
Reference Beam Count Rate (kcps)	3671.5	3645	3698
Quality Factor	4.509	4.508	4.51

## Data Availability

Plant sample from SC-CO_2_ garlic extract-treated *Triticum aestivum*. Published: Mendeley data, 17 March 2022|Version 1| DOI: 10.17632/p66v4yxbtp.1 Contributors: Barbara Kutasy, Kincső Decsi and Eszter Virág https://data.mendeley.com/datasets/p66v4yxbtp/1 (accessed on 17 March 2022). TimeCourse Table of plant sample from SC-CO_2_ garlic extract-treated *Triticum aestivum*. Published: Mendeley data, 18 March 2022|Version 1| DOI: 10.17632/xvvscxpz6w.1 Contributors: Barbara Kutasy, Kincső Decsi and Eszter Virág https://data.mendeley.com/datasets/xvvscxpz6w/1 (accessed on 18 March 2022). *Triticum aestivum* cultivar ‘Cellule’ (bread wheat). Published: National Center for Biotechnology Information (NCBI) database as BioProject PRJNA808851 (Sequence Read Archive (SRA) database: SRR18107544, SRR18107543, SRR18107542, and SRR18107541; Transcriptome Shotgun Assembly (TSA) database: GJUY00000000.1.); Registration date: 21 February 2022 Contributors: Barbara Kutasy, Kincső Decsi and Eszter Virág. https://www.ncbi.nlm.nih.gov/bioproject?LinkName=biosample_bioproject&from_uid=26113471 (accessed on 21 February 2022).

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
