# Peer review of "‘Garlic-lipo’4Plants: Liposome-Encapsulated Garlic Extract Stimulates ABA Pathway and PR Genes in Wheat (Triticum aestivum)"

_plants, 2023, doi:10.3390/plants12040743_

Round 1

Reviewer 1 Report

Overall, it is a properly designed and carried out study showing the significant effect of supercritical CO2 garlic extract encapsulated in nanoscale liposomes on wheat as estimated from the gene expression profile.

I would not say that any additional experiments or major revisions are needed. However, many minor spelling and grammatical errors could be seen throughout the manuscript - the authors should carefully proofread the text.

I would also recommend rebuilding the plots using professional statistics software such as GraphPad, SPSS, Origin or Statistica.

Some of the spelling mistakes I've found in the text are listed below.

line 27 - nutrient deficiency instead of efficiency.

34 - It should be 'by collecting samples' or 'with collecting samples

35 - spraying instead of spaying

50 - 'there was' instead of 'there were'

52 - 'reported in in vitro experiments

74 - better to say 'that they may exert a significant effect when applied'

92 - It should be 'up to 33 percents'

98 - 'and it was found that' or 'and was found to trigger the plant defence'

102 - 'profiling of genes'

Fig. 1. Better say 'Sample collection at three-time points'

131-132 - better to say 'and differentially expressed gene (DEG) analysis'

154 - please remove 'are'

165 - Better to say 'Fifty nanograms of cDNA were used'

186 - Excessive period and space after '3.1.'

213 - 'Z-score values'

222 & 225 - 'in the 24-h sample' and 'in the 48-h sample', respectively

240-241 - 'factors (TFs) that consist of HD-ZIP,...'

261 - Better say 'of the Top 50 differentially expressed genes (DEGs)'

283 - 'was' instead of 'were'(The group of genes was associated).

295-323 - Please check the punctuation and grammar in these paragraphs.

Figure 7. It makes sense to add some marks (e.g., * or #) on the graph to indicate which groups are compared.

Author Response

Dear Reviewer and Scientific Editor,                                                           28/Jan/2023

Thank you for your valuable review and comments.

Reviewer #1

Comment: “many minor spelling and grammatical errors could be seen throughout the manuscript - the authors should carefully proofread the text”

Answer: Grammatical errors and spelling were corrected and even abbreviations were checked.

Comment: “I would also recommend rebuilding the plots using professional statistics software such as GraphPad, SPSS, Origin or Statistic.”

Answer: Plots were rebuilt with the MATLAB software

Comment: “Figure 7. It makes sense to add some marks (e.g., * or #) on the graph to indicate which groups are compared.”

Answer: The plot of Real-Time-PCR (Fig. 7) has been changed and rebuilt using MATLAB, expanded with the t-test data of measurements of three replicates.

We hope that our answers are satisfactory and the corrected manuscript is acceptable for publication.

Sincerely,

Eszter Virág

Reviewer 2 Report

Manuscript plants-2180331 analyzes the effect of liposome-encapsulated garlic extract on wheat plants. The manuscript needs improvement before publication.

The material and methods Section must provide more information. More details related to the preparation of the “garlic-lipo” must be provided to allow replication of the reported works by the other scientists. The composition of the garlic extract is not defined – a brief description must be included. The garlic extract from Flavex Naturextrakte contains about 95 % olive oil. How was this oily extract encapsulated in sunflower lecithin-based liposomes? The methods must be briefly presented (for an already described method) or described in detail. Information regarding the statistical methods used from zeta potential and for results of Real-Time PCR reaction must be provided.

In the Result Section, Figure 7 must be improved. The error bar must be more prominent for a more straightforward evaluation  – please use a scale break to accommodate different values. Please use only big markers for the 15 min values–the line suggests continuing values.

In the Discussion Section.

-       L348-L356. I suggest reducing the references to socio-political requirements of low interest to scientists outside the European Union.

-       L365-L3650. It must be rewritten. In the present form, repeat the information from the results. Here it must be discussed the function of nanoliposomal formulation related to the uptake and controlled delivery of the active ingredients from garlic extract.

-       At the end of the Discussion section, the trade-off between the activation of the plant defense system and plant growth must be discussed in a couple of sentences.

In the Conclusion Section, “the immune network” must be replaced because it confuses the reader – plants do not have an immune system. I suggest replacement with a more appropriate term, plant innate immunity.

Author Response

Dear Reviewer and Scientific Editor,                                                           28/Jan/2023

Thank you for your valuable review and comments.

Please find below our answers. 

Comment: “The material and methods Section must provide more information. More details related to the preparation of the “garlic-lipo” must be provided to allow replication of the reported works by the other scientists. The composition of the garlic extract is not defined – a brief description must be included. The garlic extract from Flavex Naturextrakte contains about 95 % olive oil. How was this oily extract encapsulated in sunflower lecithin-based liposomes? The methods must be briefly presented (for an already described method) or described in detail. Information regarding the statistical methods used from zeta potential must be provided.”

Answer: We tested the ‘Garlic lipo’ agent as an active component of a commercially available product ELICE Vakcina®, a member of the Elice16 product line developed in the Research Institute for Medicinal Plants and Herbs (RIMPH) Ltd. Hungary (https://gynki.hu/en/rimph-botanicals/products/) and whose formulation and biological effect has already been characterized (see references in ms. Hegedus et al., 2022; Decsi et al., 2023). The formulation protocol is subject to the permission of the RIMPH Ltd, the owner and manufacturer of ELICE Vakcina, who cannot yet publish it due to the patent process. We have completed the ms. with everything about this agent, that the manufacturer has approved.

Information regarding the statistical methods used in zeta potential measurements were provided in the ms.

Comment: “Information regarding the statistical methods used for results of Real-Time PCR reaction must be provided. In the Result Section, Figure 7 must be improved. The error bar must be more prominent for a more straightforward evaluation – please use a scale break to accommodate different values. Please use only big markers for the 15 min values–the line suggests continuing values.”

Answer: The plot of Real-Time-PCR (Fig. 7) has been changed and expanded with the significance test of three replicates comparing data of the different times.

Comment: “In the Discussion Section. L348-L356. I suggest reducing the references to socio-political requirements of low interest to scientists outside the European Union.”

Answer:  The paragraph on agricultural policy in European Union was deleted. So this text focuses on generally the scientific community and commercial enterprises.

Comment: “L365-L3650. It must be rewritten. In the present form, repeat the information from the results. Here it must be discussed the function of nanoliposomal formulation related to the uptake and controlled delivery of the active ingredients from garlic extract.”

Answer: This is a very interesting proposal. Unfortunately, we did not perform such investigations in this work. What we know about this area so far, was described in the introduction. L90-L96. Our further experiments regarding this topic are in progress. We hope to publish that soon.

Comment: “At the end of the Discussion section, the trade-off between the activation of the plant defense system and plant growth must be discussed in a couple of sentences.”

Answer: At the end of the discussion, we added a summary section about connections between the results of the applied agent  and the plant's defense system focusing on plant physiological processes. This section was completed with an additional figure (Figure 8) that summarizes the main possible plant physiological effects of the agent based on KEGG pathway analysis.

Comment: “In the Conclusion Section, “the immune network” must be replaced because it confuses the reader – plants do not have an immune system. I suggest replacement with a more appropriate term, plant innate immunity.”

Answer:  The “immune network” phrase was changed to the plant's innate immunity and defense response mechanisms.

We hope that our answers are satisfactory and the corrected manuscript is acceptable for publication.

Sincerely,

Eszter Virág
